# Lhcx proteins provide photoprotection via thermal dissipation of absorbed light in the diatom *Phaeodactylum tricornutum*

Jochen M. Buck[1,7], Jonathan Sherman [2,7], Carolina Río Bártulos[1], Manuel Serif [1], Marc Halder[1], Jan Henkel[1,3], Angela Falciatore[4], Johann Lavaud[5], Maxim Y. Gorbunov[2], Peter G. Kroth [1], Paul G. Falkowski[2] & Bernard Lepetit [1,6]

Diatoms possess an impressive capacity for rapidly inducible thermal dissipation of excess absorbed energy (qE), provided by the xanthophyll diatoxanthin and Lhcx proteins. By knocking out the *Lhcx1* and *Lhcx2* genes individually in *Phaeodactylum tricornutum* strain 4 and complementing the knockout lines with different Lhcx proteins, multiple mutants with varying qE capacities are obtained, ranging from zero to high values. We demonstrate that qE is entirely dependent on the concerted action of diatoxanthin and Lhcx proteins, with Lhcx1, Lhcx2 and Lhcx3 having similar functions. Moreover, we establish a clear link between Lhcx1/2/3 mediated inducible thermal energy dissipation and a reduction in the functional absorption cross-section of photosystem II. This regulation of the functional absorption cross-section can be tuned by altered Lhcx protein expression in response to environmental conditions. Our results provide a holistic understanding of the rapidly inducible thermal energy dissipation process and its mechanistic implications in diatoms.

[1] Plant Ecophysiology, Department of Biology, University of Konstanz, 78457 Konstanz, Germany. [2] Environmental Biophysics and Molecular Ecology Program, Department of Marine and Coastal Sciences, Rutgers, The State University of New Jersey, New Brunswick, NJ 08901, USA. [3] Institute of Genetics, Vetsuisse Faculty, University of Bern, 3001 Bern, Switzerland. [4] Sorbonne Université, Centre National de la Recherche Scientifique, Institut de Biologie Paris-Seine, Laboratory of Computational and Quantitative Biology, F-75005 Paris, France. [5] UMI 3376 Takuvik, CNRS/ULaval, Département de Biologie, Pavillon Alexandre-Vachon, Université Laval, Québec (Québec) G1V 0A6, Canada. [6] Zukunftskolleg, University of Konstanz, 78457 Konstanz, Germany. [7] These authors contributed equally: Jochen M. Buck, Jonathan Sherman. Correspondence and requests for materials should be addressed to B.L. (email: bernard.lepetit@uni-konstanz.de)

Upon absorption of a photon, the singlet excited state of a chlorophyll *a* molecule has three major fates: photochemistry, heat dissipation, and fluorescence emission. The optimization of photochemistry is achieved when the rate of photon absorption is equal to the rate of electron transfer. Although this requires a large absorption cross-section of photosystem II under low light (LL) conditions, the same absorption cross-section would lead to an overflow of energy in the photosynthetic system under high light conditions, resulting in massive oxidative damage. Hence, in oxygenic photosynthetic organisms, photon capture can be regulated by adjusting the cross-section of photosystem II on time scales of minutes. In practice, this phenomenon leads to changes in the balance between excitons directed to reaction centers and those that are dissipated as heat.

There are a number of photoprotective mechanisms that thermally dissipate excess absorbed energy as heat; collectively these are called Non-Photochemical Quenching (NPQ). This phenomenon is present in all photosynthetic eukaryotes and in many cyanobacteria[1–3], and is characterized by a downregulation of chlorophyll fluorescence at high irradiance. Mechanistically, NPQ comprises several processes[3,4]. The most-rapid NPQ component, called energy-dependent quenching (qE), is strongly dependent on light intensity, reflecting the balance between fluorescence quenching and an increase in thermal dissipation[5]. qE has been shown to decrease the functional absorption cross-section of PSII (σPSII), thus reducing the flux of absorbed energy into photochemistry[5–10]. Consequently, light-enhanced thermal dissipation reduces the excitation pressure and hence the probability of photooxidative damage. In higher plants, however, the correlation between the onset of qE and the reduction of σPSII has recently been challenged[11].

qE has been found in several major algal lineages as well as in mosses and land plants[3,12]. The initiation of qE is driven by the establishment of a pH gradient between the thylakoid lumen and plastid stroma (ΔpH) upon excess light exposure. It often also correlates with the conversion of xanthophyll pigments in the so-called xanthophyll cycle, which, for diatoms, is the conversion of diadinoxanthin (Dd) into diatoxanthin (Dt)[13–15]. In addition, in plants, qE requires the presence of PsbS, a specialized protein that belongs to the light harvesting complex protein (LHC) family[16]. In contrast, the green alga *Chlamydomonas reinhardtii* requires Lhcsr proteins for qE[17]. Similarly, in diatoms, the importance of the Lhcsr-related Lhcx1 proteins for qE has been proven by silencing the *Lhcx1* genes in *Phaeodactylum tricornutum*[18] and *Cyclotella meneghiniana*[19].

Diatoms are unicellular microalgae with complex plastids, which were acquired during serial secondary endosymbiosis from a green and a red alga[20]. They constitute one of the most important phytoplankton groups in terms of productivity[21] and biodiversity[22]. Their abundance is also based on their capacity to thrive in turbulent waters (e.g., coasts or upwelling regions), where they can exploit the huge amount of available nutrients[23]. This goes in line with their variable qE capacity depending on the light characteristics of the respective habitats[24–26]. In contrast to *C. reinhardtii*, which contains only two different Lhcsr proteins, many diatoms possess multiple Lhcx proteins, e.g., *Fragilariopsis cylindrus* contains 11 Lhcx[27]. For several diatom species, qE capacity correlates with the expression of different Lhcx proteins, which indicates the involvement of various Lhcx proteins in triggering qE under different environmental conditions[28–35]. *P. tricornutum* contains only four Lhcx proteins[36] and therefore is a particularly good model to study the impact of individual Lhcx proteins on qE capacity. These four proteins are structurally similar, but are differentially expressed under varying environmental conditions[32].

Here, we use a model diatom, *P. tricornutum*, to examine two basic questions: First, besides Lhcx1, do other Lhcx proteins equally confer qE capacity and if so, is it correlated with the xanthophyll cycle. Second, is qE in diatoms correlated with a reduction of σPSII. Experimentally, we knockout the *Lhcx1* gene in *P. tricornutum* that results in mutant lines devoid of qE. Then, we individually overexpress all other Lhcx proteins in this knockout background. In addition, we knockout the *Lhcx2* gene and express the Lhcx2 and Lhcx3 protein in this knockout background. This experimental design allows us to quantify the influence of each Lhcx protein on qE and σPSII.

## Results

**The effect of different Lhcx proteins on qE.** By using the TALEN method[37], we independently targeted two different sites of the *Lhcx1* gene in *P. tricornutum* strain Pt4 (UTEX 646). Three Lhcx1 knockout (x1KO) lines were obtained (cf. Supplementary Fig. 1, Supplementary Fig. 2 for genetic characterization via PCR and 125 bp paired end whole genome sequencing), which lacked the Lhcx1 protein under LL growth conditions (Fig. 1a). Two of these are based on TALEN pair 1 (x1KO_1a/1b), and the third on TALEN pair 2 (x1KO_2). Although the wild-type cells activated qE rapidly during 3 mins of supra-optimal light exposure, the three x1KO clones lacked this qE induction (Fig. 1b). Moreover, the very low NPQ values obtained after 3 min of supra-optimal light exposure in the x1KO lines even increased during the following dark phase, indicating it is not of qE origin. To prove that the observed phenotype is indeed related to the knockout of Lhcx1, we complemented the independent x1KO_1a and x1KO_2 lines with an *Lhcx1* gene that was modified at the TALEN-binding sites by synonymous codon usage, in order to prevent a re-cutting by the TALEN system. To express the gene, we used either the native *Lhcx1* promoter/terminator or the *FcpA* (*Lhcf1*) promoter/terminator. Out of several lines created (Supplementary Fig. 1a and b), some were further characterized regarding their NPQ characteristics. These lines showed a rescue of the qE phenotype to different extents (Fig. 1b).

For closer examination, we used the x1KO_1a strain, as it showed no statistically significant difference in growth compared with the wild-type (Fig. 1c), which is expected for cells cultivated under LL conditions where qE is not induced. We transformed this strain with each of the three other *Lhcx* genes to generate x1KO + x2/x3/x4-supplemented strains. To ensure similar regulation and expression of the other *Lhcx* genes as the original *Lhcx1*, we used the *Lhcx1* promoter and terminator in all transformed strains. Our aim was to see the possible effects of each Lhcx protein on triggering qE. Normally, these effects are hidden or dampened by the presence of Lhcx1, which is the most-expressed Lhcx protein under LL cultivation[32] (Fig. 1a). From the obtained clones, we chose two for the *Lhcx2* and *Lhcx3*, respectively, and three for the *Lhcx4* gene, all of which showed strong expression of the respective gene (Supplementary Fig. 3) as well as protein under LL growth conditions (Fig. 2). We then investigated the qE pattern upon exposure to 10 min of supra-optimal actinic light and subsequent recovery under LL conditions known to relax qE in *P. tricornutum* better than darkness[14]. Although the x1KO line showed only a slight linear increase in NPQ, which did not relax under LL conditions and may rather be related to photoinhibition processes (qI), the x1KO + x2 and x1KO + x3-supplemented lines recovered qE capacity (Fig. 3a; Supplementary Fig. 4). The extent of qE varied in the two chosen strains supplemented with the same gene (i.e., x2a vs. x2b, and x3a vs. x3b; Fig. 3a; Supplementary Fig. 4), most likely caused by differential expression of the respective genes

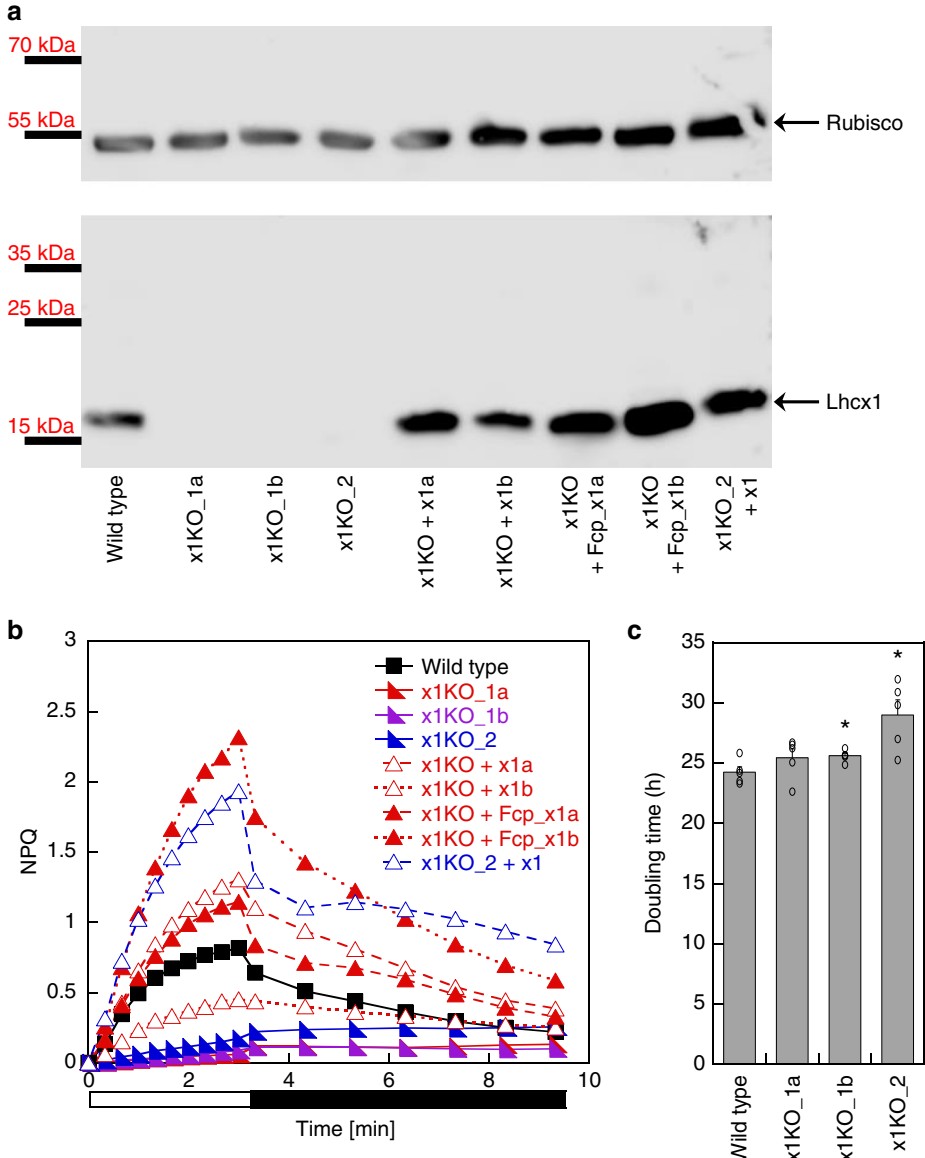

**Fig. 1** qE capacity in *P. tricornutum* Lhcx1 mutants. **a** Western blot of low light grown strains, from left to right: wild type, three x1KO lines (x1KO_1a,1b and 2), two complemented lines of the x1KO_1a with the *Lhcx1* gene cloned between the *Lhcx1* promoter and terminator (x1KO + x1a,b) and between the *FcpA* promoter (x1KO + Fcp_x1a,b), respectively, and one complemented line of the x1KO_2 line (x1KO_2 + x1). After blotting, the blot was cut and the upper part was incubated with a Rubisco antibody, whereas the lower part was incubated with the Lhcx antibody. Protein mass marker bands are indicated on the left. **b** NPQ development during 3 min of actinic high light (white bar) followed by 6 min of recovery in darkness (black bar). Representative traces are depicted. **c** Mean doubling time in the exponential growth phase in the wild type and three x1KO lines during low light growth. SE is given. Statistically significant differences between the x1KO lines and the wild-type are indicated by a * (unpaired *t* test, eight degrees of freedom, $p < 0.05$, $n = 5$ biological replicates). Source data are provided as a Source Data file

owing to positional effects on the inserted vector and the amount of inserted copies[38]. Interestingly, Lhcx4-supplemented lines were unable to restore qE capacity (Fig. 3a; Supplementary Fig. 4).

Diatoms show a strong correlation between qE and the concentration of Dt, which is formed via de-epoxidation of Dd[13–15,31,39,40]. However, Dd de-epoxidation was similar in all our strains regardless of qE capacity (Fig. 3b; Supplementary Fig. 5). When the Dd to Dt conversion was inhibited by dithiothreitol (DTT) in the qE containing strains (wild-type, x1KO + x2, x1KO + x3), the qE capacity was lost (Fig. 3; Supplementary Fig. 4). Instead, those strains exhibited NPQ characteristics similar to the x1KO mutants without DTT treatment. Hence, Dt can only confer qE in the presence of Lhcx1/2/3 proteins, and vice

versa, and the slower NPQ phase, observed under prolonged high light intensities, is independent of both compounds.

Under LL conditions, *Lhcx1* is the highest expressed *Lhcx* gene, followed by *Lhcx2*, whereas *Lhcx3* and *Lhcx4* are hardly expressed[32]. We created an Lhcx2-KO line (x2KO, verified by allele specific PCR and sequencing, qPCR and western blot, Supplementary Fig. 6, 7, and 8), which did not exhibit a reduction in qE capacity under LL cultivation (Inset Supplementary Fig. 9). However, Lhcx2 had been proposed to provide additional qE capacity under prolonged exposure to supra-optimal light[30,33,34]. Indeed, the x2KO strain, which lacks the high light induction of Lhcx2 (Supplementary Fig. 7 and 8), showed a lower increase of qE capacity compared with the wild-type when exposed to

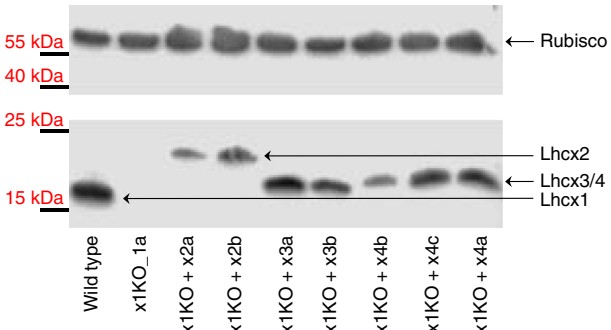

**Fig. 2** Western blot of wild type, x1KO_1a and supplemented x1KO_1a strains. Cells were grown under low light. After blotting, the blot was cut and the upper part was incubated with a Rubisco antibody, whereas the lower part was incubated with the Lhcx antibody. Lhcx1 has the lowest molecular mass, Lhcx2 the highest, and Lhcx3 and Lhcx4 have a molecular mass in between. Protein mass marker bands are indicated on the left. Source data are provided as a Source Data file

2 hours of supra-optimal light (1700 μmol photons $m^{-2}$ $s^{-1}$), particularly during the second hour (Supplementary Fig. 9). Still, the clearly visible increase of qE is likely caused by the upregulation of Lhcx3 (Supplementary Fig. 8)[33,34]. The x2KO + x2-complemented line rescued the capacity for qE increase during prolonged supra-optimal illumination and displayed the highest capacity for qE right from the beginning.

**The relationship between qE and σPSII.** The different *Lhcx*-mutated lines generated offered us a unique opportunity to investigate the extent to which Lhcx-mediated qE can influence σPSII in diatoms. σPSII, the functional absorption cross-section of PSII, represents the probability of an absorbed photon of a given wavelength to drive a successful charge separation. To examine this phenomenon, we selected those strains that exhibited varying degrees of qE when grown under LL conditions. For reference, we ordered these lines from high to no qE capacity: x2KO + x2 > wild-type = x2KO > x1KO + x3a > x1KO + x2a > x1KO = x1KO + x4a. In addition, we included another strain, in which Lhcx3 is expressed under the control of the *Lhcx2* promoter in the x2KO background, termed x2KO + x3. This strain had a similar qE capacity as the wild-type and the x2KO line under LL cultivation. For these strains we measured variable fluorescence as a function of light intensity at 15 light steps. After each light step, cells were exposed to a short flash leading to a single turnover of PSII, hence the transfer of one electron from the PSII reaction center to the acceptor $Q_A$, from which electron transport rates and σPSII can be calculated, the latter by fitting the rise in fluorescence to a cumulative one-hit Poisson function[41]. Electron transport rates started to saturate at ~ 130 μmol photons $m^{-2}$ $s^{-1}$ for all cultures (Supplementary Fig. 10). qE became apparent at similar light intensities in wild-type, x1KO + x3a, x2KO, x2KO + x2, and x2KO + x3. Following the initiation, qE rapidly increased until light intensities of ~ 350 μmol photons $m^{-2}$ $s^{-1}$, after which NPQ rose slowly until reaching its maximum values of 0.6–0.7 at 800 μmol photons $m^{-2}$ $s^{-1}$; in the x2KO + x2 line NPQ reached even 0.8 (Supplementary Fig. 11; Fig. 4a). In contrast, in the x1KO and x1KO + x4a strains NPQ decreased to negative values at light intensities up to 250 μmol photons $m^{-2}$ $s^{-1}$ and reached maximum values of ~ 0.2 at 800 μmol photons $m^{-2}$ $s^{-1}$. In the x1KO + x2a strain, qE started at a higher light intensity in comparison to the wild-type, and maximum NPQ was in between the x1KO/x1KO + x4a and the other strains. When inhibiting Dt synthesis with DTT, the rapid phase of NPQ development in the qE-possessing strains was completely abolished, resulting in NPQ traces resembling those

of the x1KO and x1KO + x4a strains without DTT (Supplementary Fig. 11). In the two latter strains, DTT application did not change NPQ characteristics at all. We do not consider the slight linear increase of NPQ at higher light intensities as qE. We observed this increase in all strains and it was independent of the presence of Lhcx proteins or of a functional Dd de-epoxidation.

σPSII values of low light grown strains, measured in the dark, were between 500 and 550 $Å^2$ $PSII^{-1}$ upon blue light (450 nm) exposure, typical for *P. tricornutum*[42]. We observed a slight decline in σPSII as light intensities increased to 70–100 μmol photons $m^{-2}$ $s^{-1}$ (Fig. 4a; Supplementary Fig. 12). At higher light intensities up to 350–400 μmol photons $m^{-2}$ $s^{-1}$, σPSII increased in the x1KO and the x1KO + x4a lines and remained at the same level in x1KO + x2a line. In contrast, σPSII decreased to a greater extent in the strains with high qE capacity (wild-type, x1KO + x3a, x2KO, x2KO + x2/x3), down to values of 400–450 $Å^2$ $PSII^{-1}$ (a reduction of 15–20%). Further increased light intensities, which induced a slight linear increase of NPQ capacity not related to qE (see above), did not further downregulate σPSII. When inhibiting Dt synthesis by addition of DTT, qE possessing strains showed a σPSII development similar to the x1KO and the x1KO + x4a strains where σPSII was unaffected by the addition of DTT (Supplementary Fig. 12). Besides calculating the changes in σPSII based on single turnover saturating flashes, leading to one $Q_A^-$ per PSII, σPSII is also often determined by applying prolonged weak flashes in the presence of DCMU, which yields to a full reduction of $Q_A$, but has some side effects due to the application of DCMU (reviewed in ref. 10). In order to corroborate our results, we compared the effect of three minutes supra-optimal light exposure on NPQ establishment and σPSII behavior in wild-type and x1KO strains using the DCMU method and applying the calculation and correction procedure as described by Tian et al.[10]. Using this method, we also observed a significant reduction in σPSII in the wild type after supra-optimal light exposure, which was absent in the x1KO strain (Supplementary Fig. 13 and 14). Based on these results, we conclude that there is a pronounced influence of qE on changes in σPSII.

Furthermore, we investigated the relationship between σPSII and NPQ after 1 day of high light growth. qE capacity increased in strains, which already possessed qE capacity under LL growth and was now present even in the x1KO and x1KO + x4a line (Fig. 4b). This is owing to the high amounts of Lhcx2 and Lhcx3 (cf. protein levels under LL and high light cultivation, Fig. 2, Supplementary Fig. 8), which can partially rescue qE capacity in the absence of Lhcx1, as already demonstrated by the analysis of the x1KO supplemented lines (cf. Fig. 3). As opposed to LL grown cells, qE initiated below 100 μmol photons $m^{-2}$ $s^{-1}$ and leveled off at 300–400 μmol photons $m^{-2}$ $s^{-1}$, thereafter only slowly rising with higher light intensities. The highest NPQ values were obtained in the x2KO + x2/x3 and the x1KO + x3a lines, with NPQ values of 1.6–1.7 at 400 μmol photons $m^{-2}$ $s^{-1}$, i.e., when the rapid phase of qE induction was complete. Simultaneously, σPSII decreased in all lines substantially, by up to 40–45% in wild-type, x1KO + x3a, x2KO, and x2KO + x2/x3 and by up to 30% in x1KO and x1KO + x4a (Fig. 4b; Supplementary Fig. 12). Strain x1KO + x2a displayed a decrease in between these two groups. As for LL grown cells, the rapid decline of σPSII up to light intensities of 250–400 μmol photons $m^{-2}$ $s^{-1}$ well correlated with the rapid development of qE, with the exception of the decline of σPSII under very weak light intensities in the beginning (Fig. 4b). The weaker and linear increase of NPQ under further increased light intensities did not have any effect on σPSII.

We also investigated the excitation pressure on PSII (monitored as 1-qL, a proxy for the reduction state of the plastoquinone pool[43]) required to reduce σPSII. Strains possessing a higher qE capacity exhibited a substantial decrease of σPSII

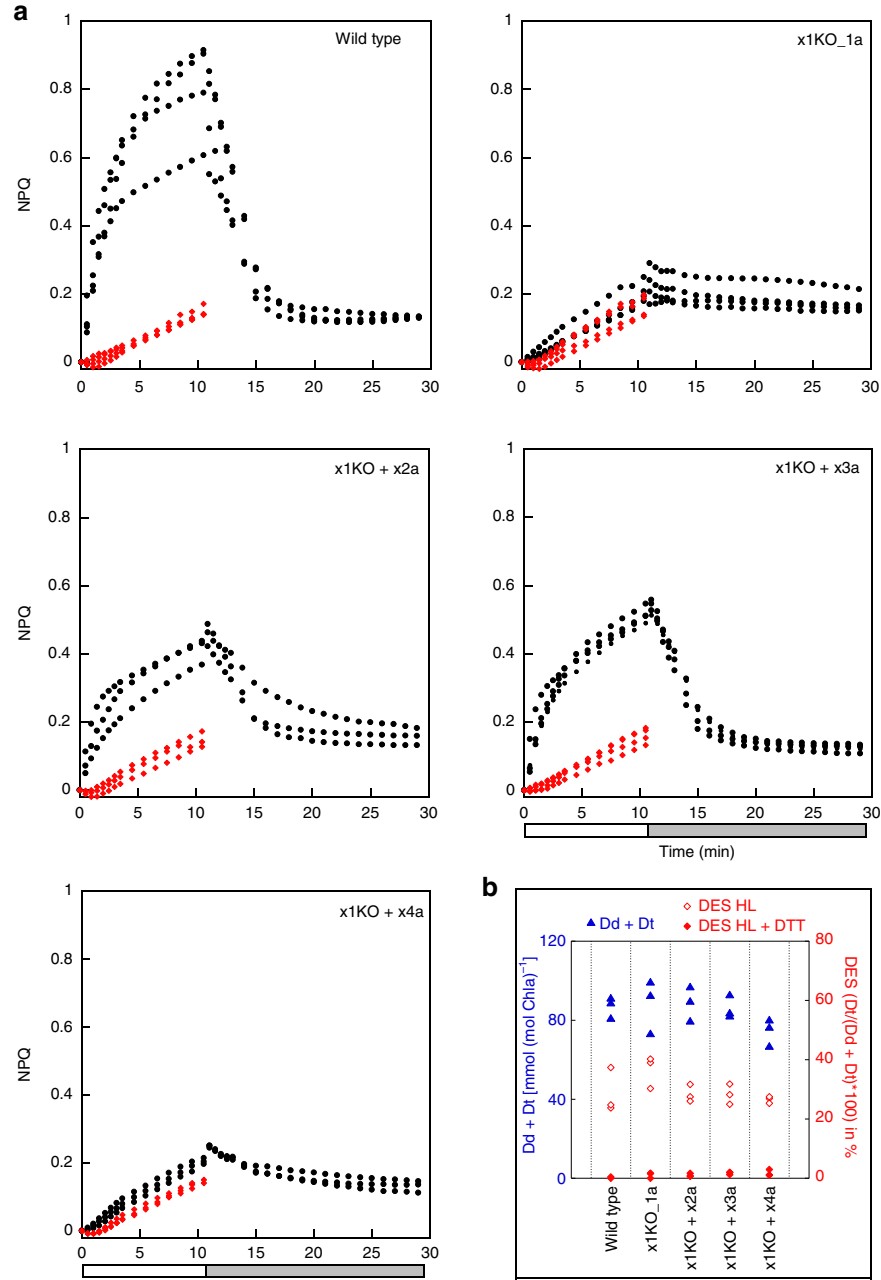

**Fig. 3** NPQ kinetics and xanthophyll cycle activity in wild type and mutant strains. Strains were concentrated to a chlorophyll $a$ amount of 10 mg L$^{-1}$. **a** NPQ capacity of wild type (four biological replicates (BR)), x1KO_1a (four BR), and x1KO_1a supplemented lines x1KO + x2a (three BR), x1KO + x3a (four BR) and x1KO + x4a (three BR) during 10 min exposure to 1700 µmol photons m$^{-2}$ s$^{-1}$ (white bar), followed by 18 min of low light (gray bar). Red points indicate samples that had been incubated with DTT prior to high light exposure in order to prevent diatoxanthin formation; **b** pool size of diadinoxanthin + diatoxanthin (Dd + Dt) per chlorophyll $a$ and de-epoxidation state (DES) following 10 min illumination with 1700 µmol photons m$^{-2}$ s$^{-1}$ without and with prior application of DTT. Three BR were measured. Source data are provided as a Source Data file

at much lower PSII excitation pressure. While for LL grown cultures 1-qL values of 0.6–0.8 were necessary to reduce σPSII in qE possessing strains, high light-acclimated cultures started to decrease σPSII at 1-qL values of ~ 0.2–0.3 (Supplementary Fig. 15). This correlates with the onset of qE at lower light intensities in high light vs. LL grown cultures (cf. Fig. 4).

NPQ is the most widespread parameter used to characterize thermal dissipation. However, owing to its derivation method, it may exaggerate the effect of thermal dissipation at higher values. To bound NPQ between 0 and 1, the parameter Y(NPQ) has been proposed to better visualize the effect of thermal dissipation[9,44,45].

By plotting NPQ versus Y(NPQ) we observed an almost linear correlation up to NPQ values of ~ 0.6 (Supplementary Fig. 16). Above this threshold, the relative increase of NPQ vs Y(NPQ) was enhanced. The maximum Y(NPQ) values were obtained for high light grown cultures and reached ~ 0.6, i.e., an induced thermal dissipation of 60% of absorbed photons. We plotted all σPSII values against Y(NPQ) for both LL and high light grown strains together, excluding those points where Y(NPQ) was absent or did not further downregulate σPSII (see details legend Fig. 5). The latter points represent σPSII values measured when NPQ shifted from the fast qE component to a slower component and had no

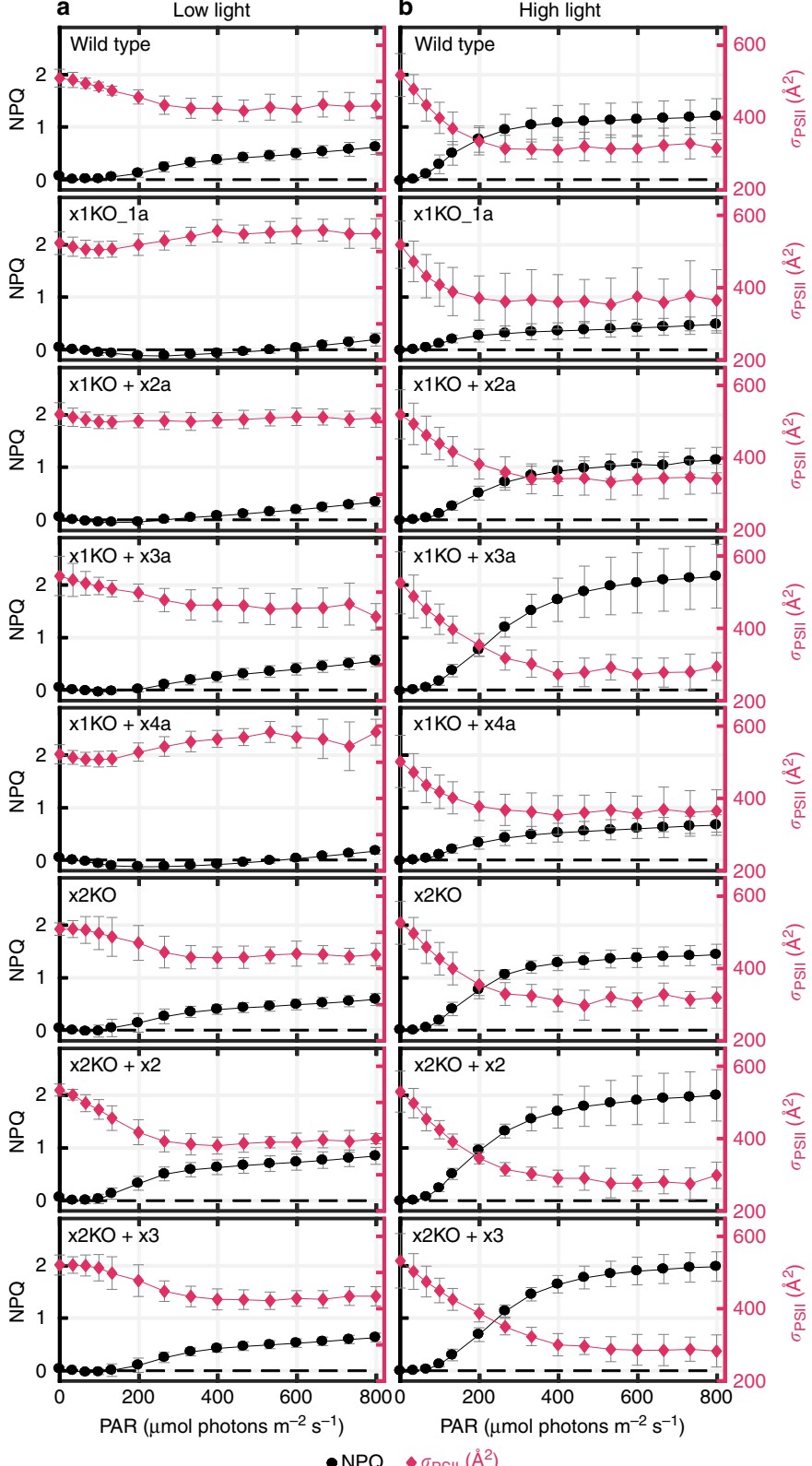

**Fig. 4** NPQ development and changes in σPSII during increasing light intensities. Wild-type and mutant strains cultivated under low light (**a**) and 24 h of high light ( ~ 400 μmol photons m$^{-2}$ s$^{-1}$) (**b**) were exposed to 15 steps of increasing light intensity (1 min duration each), and NPQ (black trace) and σPSII (red trace) were recorded. Values are the mean of six (low light cultures) or five (high light cultures) biological replicates. SD is given. Dashed black line denotes zero NPQ. Source data are provided as a Source Data file

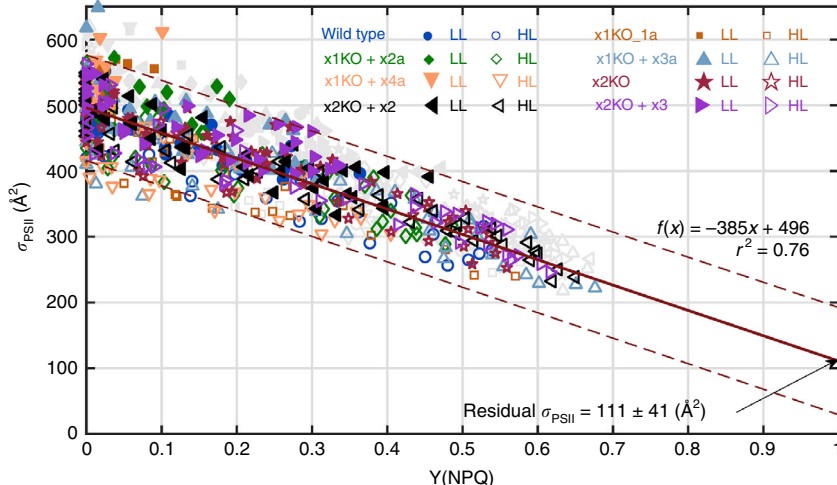

**Fig. 5** σPSII vs Y(NPQ) from rapid light curves and the corresponding linear regression. Individual data points of all measured strains cultivated both under low light (six biological replicates each) and high light (five biological replicates each). In order to discern Lhcx and Dt-dependent qE processes from qI processes, data points where an increase of Y(NPQ) did not lead to a further downregulation of σPSII are not included in the regression calculation, but indicated in light gray. This was determined by calculating the percent change of each σPSII from its previous light step. If σPSII decreased by < 5% of the total measured decrease for that curve while Y(NPQ) increased, it was omitted. Data points above 600 μmol photons $m^{-2}$ $s^{-1}$ and data points with a negative Y(NPQ) were also removed. A linear regression line, the 95% confidence interval, the regression equation and the $r^2$ are indicated. Source data are provided as a Source Data file

additional effect on σPSII (see discussion). Interestingly, we obtained a fairly good linear correlation between σPSII and Y(NPQ). This indicates that no matter which Lhcx proteins are expressed, they all mediate qE in parallel to a downregulation of σPSII (Fig. 5, Supplementary Fig. 17).

**The relationship between qE and fluorescence lifetimes.** At last, we investigated the relationship between qE capacity and the average fluorescence lifetimes when qE was induced by 10 min of continuous supra-optimal illumination. Fluorescence lifetime is a measure of the time span a molecule remains in the excited state before returning to the ground state through emitting fluorescence. With the induction of qE and the resultant fluorescence quenching, fluorescence lifetimes shorten[46]. In line with the rapid onset of qE (cf. Fig. 3), we saw a pronounced decrease of fluorescence lifetimes in the LL grown qE possessing strains (wild-type, x1KO + x2a, x1KO + x3a, x2KO, x2KO + x2, x2KO + x3) during the first minute of illumination, which was absent in the x1KO and the x1KO + x4a strain (Supplementary Fig. 18). Thereafter, there was a slight linear decline in all strains. In contrast, in high light grown cells a rapid decrease of fluorescence lifetimes was visible in all strains already after 1 min of illumination. Overall, the decreases in lifetime well reflected the capacity for qE in the different strains.

## Discussion

By knocking out the *Lhcx1* gene, we obtained the *P. tricornutum* strain (x1KO) that had no capacity for qE, i.e., the rapid component of NPQ (Fig. 3; Fig. 4). Our results support previous findings by Bailleul et al.[18] regarding the involvement of Lhcx1 in qE. We suggest that Lhcx1 provides essentially all qE capacity in *P. tricornutum* grown under LL conditions. This is different to *C. reinhardtii*, where no qE is developed under LL growth conditions[17], but both existing Lhcsr proteins provide qE after prolonged exposure to high light[47]. Our results also revealed that Lhcx2 and Lhcx3 provide qE capacity in *P. tricornutum*, alongside Lhcx1, but cannot specify a role for Lhcx4 in this process, at least not for high light inducible, Dt-dependent qE and independent of

Lhcx1. This is in line with the strong upregulation of Lhcx4 under prolonged dark conditions[32,48] which requires future research to elucidate its function.

As a consequence of the lack of Lhcx1, the x1KO strain, as well as the x1KO + x4a strain, exhibit no decrease in σPSII under high light exposure. Instead, they show a slight increase in σPSII, as expected when PSII reaction centers become progressively closed, causing excitons to jump from one PSII unit to another, in search for an open PSII center[49,50]. On the other hand, the onset of qE in wild-type and the five other strains leads to a substantial decrease in both fluorescence lifetimes and σPSII. This decrease in σPSII clearly indicates a reduction in the flux of energy to the PSII core, which provides photoprotection under supra-optimal light. While previously it has been shown that the decrease in σPSII correlates to a certain degree with the NPQ capacity in different algal species[8–10,13,46,51,52], here, we prove that in the diatom *P. tricornutum* this decrease is mediated in a concerted manner via Lhcx proteins (Lhcx1, Lhcx2, and Lhcx3) and the conversion of Dd to Dt. Lhcx proteins or the xanthophyll cycle alone—at least under LL growth conditions - do not activate qE or cause a reduction in σPSII under supra-optimal irradiances.

As Dt binds to LHC antenna proteins[53,54] and Lhcx proteins are not part of the PSII core[55], the decrease in σPSII is indicative of thermal dissipation of absorbed energy in the LHC antenna complexes, as originally proposed by Genty et al.[5]. Accordingly, analysis of picosecond lifetime kinetics revealed that the exposure to supra-optimal irradiances leads to functional modifications in both antennae and PSII reaction centers, but the thermal dissipation occurs only in the antennae[46]. Our results also showed that the slow phase of NPQ, which is activated under prolonged exposure to higher light levels, has no effect on σPSII. A similar slow NPQ phase with no reduction in σPSII is observed in the mutants lacking qE (x1KO and x1KO + x4) as well as in all strains treated with DTT (which inhibits the formation of Dt). These results suggest that the slow phase of NPQ is related to photoinhibitory damage to the PSII reaction center (qI).

The combination of NPQ and σPSII measurements in a variety of mutants with different NPQ extents allowed us to better characterize multiple NPQ processes that are otherwise hardly

distinguishable[3,4]. Accordingly, the most rapid NPQ process (qE) can be easily identified by a pronounced reduction in σPSII upon exposure to high light. Whether this scenario holds true also for other algal taxa remains to be tested, but recent results in the green alga *Ostreococcus taurii* suggest a similar correlation between qE and σPSII[9]. Our measurements of a significantly decreased σPSII under qE conditions are fully consistent with the proposal that qE shortens the time an exciton can travel before being thermally dissipated, and thus decreases the effective excitation diffusion length for an exciton to reach an open PSII reaction center[56].

Because the PSII reaction center is energetically a shallow trap[46,57], excitons that reach a PSII core may transfer back to the peripheral antennae where they can be thermally dissipated once all PSII reaction centers are closed. Such a mechanism has been identified in plants, termed "economic photoprotection"[11], and later was also postulated to exist in diatoms[51]. This economic photoprotection, however, does not involve a decrease in σPSII upon induction of qE[11], which is in sharp contrast to our results.

As we obtained a linear correlation between the extent of Y (NPQ) and the decrease in σPSII driven directly by qE (Fig. 5, Supplementary Fig. 17), we can use the regression equation to calculate σPSII for a theoretical maximal Y(NPQ) of 1, i.e., where all absorbed energy is dissipated as heat. This leads to a value of σPSII = 111 Å$^2$ at Y(NPQ) = 1 (and 115 Å$^2$ if both parameters are measured after 1 s darkness). Using a chlorophyll *a* specific absorption coefficient of 9.8 m$^2$ (g chlorophyll *a*)$^{-1}$ for blue light[58] (as used here for the σPSII measurements) and assuming a dimeric PSII core for diatoms in vivo[59], containing 70 chlorophyll *a* molecules[55], we calculated a functional PSII core cross-section of 112 Å$^2$. This value is virtually identical to the one calculated at Y(NPQ) = 1 based on our σPSII vs. Y(NPQ) regression. In a similar approach for the centric diatom *T. pseudonana*, Campbell, and co-workers calculated with a monomeric PSII and concluded that at Y(NPQ) = 1 the remaining σPSII is provided by the PSII core and some LHCs that are not thermally downregulated[9]. If one assumes that the existing dimeric PSII[59] is excitonically coupled and shares a common peripheral LHC antenna, then the regression to a Y(NPQ) of 1 provides a residual σPSII, which also in *T. pseudonana* corresponds to the functional absorption cross-section of the dimeric PSII core. In either cases, the results of Xu et al.[9] and this study indicate that regulation of σPSII upon induction of qE does not involve the PSII core, otherwise the regression at Y(NPQ) of 1 would lead to σPSII values close to 0. The most parsimonious explanation is that Lhcx-mediated qE mechanistically leads to a disconnection of the peripheral LHC antennae—very much in line with recent results obtained by a different experimental approach[51]—and these functionally disconnected LHC antennae dissipate the excess absorbed light as heat. Such a mechanism has been identified as one compound of NPQ in *Arabidopsis thaliana* using electron microscopy[60].

It is not clear yet how this disconnection of the LHC antennae is achieved, but recent results indicate a strong role of Dt molecules in influencing thylakoid membrane rigidity[61]. Moreover, traces of Lhcx proteins have been detected in photosystem II preparations[34], indicating that Lhcx proteins might be more strongly connected to the PSII core than the peripheral LHC proteins. Both aspects could influence peripheral LHC antennae connectivity with the PSII cores in such a way that, upon induction of qE, the peripheral antennae are moved away slightly, exceeding the maximal distance for functional Förster resonance energy transfer to the PSII core. In such functionally disconnected LHC antennae, internal heat dissipation and chlorophyll fluorescence emission—the remaining energy dissipation pathways—would compete and a further synthesis of Dt under prolonged supra-optimal light would decrease fluorescence yield even more

by increasing thermal dissipation, but without further affecting σPSII. These processes, together with photoinhibited PSII reaction centers[46], may explain the observation of very high NPQ values, which do not affect σPSII[8,9,51].

There had been several reports on two potential qE quenching loci in diatoms based on fluorescence lifetime kinetics. The first is associated with an uncoupling of antenna complexes from the PSII core, which is independent of Dt formation. The second is taking place close to the PSII reaction center and is directly dependent on the xanthophyll cycle supposedly[3,25,34,51,62,63]. These reports are corroborated by classical PAM analyses in centric diatoms, where one component of qE is independent of the conversion of Dd to Dt[40] and clearly different antenna organizations exist[64]. However, the two qE-quenching sites concept is difficult to reconcile with the prime dependency of qE on the amount of Dt in the pennate *P. tricornutum*[13–15,39] and with our results, which point to only one mechanism of qE. One reason for this discrepancy may be that in both time resolved fluorescence studies revealing the two quenching sites in *P. tricornutum*[34,62] the time span of cells exposed to supra-optimal light conditions was much longer than necessary to induce full qE and hence also qI related processes likely had been recorded. Indeed, when Taddei et al.[34] and colleagues measured LL grown *P. tricornutum* cells, they observed only one quenching site. Analyses of picosecond lifetime kinetics during the NPQ development in *P. tricornutum* revealed a functional modification in the PSII reaction center, in line with previous results[65], which, however, is independent of the xanthophyll cycle and does not lead to any thermal dissipation[46]. Instead, this modification in the energy transfer in PSII centers reduces the probability of producing triplet chlorophyll[46]. Our results demonstrate that only Dt- and Lhcx-mediated qE provides thermal dissipation in the antennae, which likely involves uncoupling of functional antennae. This is characterized by one quenching site that corroborates the conclusions obtained by Kuzminov and Gorbunov[46].

We have to note that our analyses have been conducted in *P. tricornutum* strain 4, which had been originally isolated from the Baltic sea and which has a lower NPQ capacity than other *P. tricornutum* strains[18]. Some analyses regarding two different quenching sites and mechanisms had been performed in another *P. tricornutum* strain (Pt2)[51,62], which has a higher NPQ capacity than Pt4. Thus, in principle there could exist an additional quenching mechanism in Pt2. However, the fact that also in Pt2 there is a 1:1 dependency of NPQ on the amount of Dt[39], together with the explanations for the apparent occurrence of a second quenching site provided above, rather argue against this assumption.

Finally, it had been speculated that different Lhcx proteins have different capacities to provide NPQ, with Lhcx1 being the most effective[34]. We indeed observed no qE in the LL grown cells if Lhcx1 is knocked out, but this is rather owing to a much lower expression of Lhcx2 and Lhcx3 under these conditions[32] (Fig. 2). Although we observe different extents of qE in the x1KO supplemented lines as compared with the wild-type, it remains unclear whether this is owing to an altered efficiency of Lhcx2 and Lhcx3 in providing qE or owing to differential protein expression. Also, whether qE provided by the different Lhcx proteins is activated at different light exposure conditions (either light intensity or exposure duration) remains to be elucidated. However, our results clearly demonstrate that the extent of qE always correlates with a reduction in σPSII with the same linear relation in all lines (Fig. 5, Supplementary Fig. 17). This implies that Lhcx1/2/3 have the same capacity for decreasing the flux of absorbed light energy to PSII via thermal dissipation in the LHC antennae. Although some individual Lhcx proteins, such as Lhcx4, could possess other functions than qE, our results suggest

that the primary role of Lhcx proteins is providing qE by decreasing the functional absorption cross-section.

Then, why do diatoms have several different *Lhcx* genes, in the case of the polar strain *Fragilariopsis cylindrus* even 11? As has been shown in[30,32,33,66,67], all four Lhcx proteins of *P. tricornutum* are modulated in their expression by different environmental triggers, such as, e.g., low light (Lhcx1), high light (Lhcx2, Lhcx3), blue light (Lhcx1-3), iron limitation (Lhcx2), or nitrogen starvation (Lhcx3, Lhcx4). We propose that one gene alone cannot contain all required regulatory cis-elements in order to respond to the multitude of environmental triggers. Instead, through gene duplications during evolution, several regulatory cis-elements could be integrated into several *Lhcx* gene promoters to modulate the expression of proteins with very similar functions and thus to tune the functional absorption cross-section depending on the environmental conditions.

Overall, our study reveals the molecular mechanism of how diatoms fine tune qE, allowing them to thrive in continuously changing light environments, where NPQ is one of the most important physiological processes[68].

## Methods

**Cell culturing**. Experiments were performed in *Phaeodactylum tricornutum* strain 4 (Pt4, UTEX 646). All strains were grown in batch cultures on a shaker at 20 °C in a 16 h day/8 h night cycle exposed to white light with an intensity of 40 μmol photons m$^{-2}$ s$^{-1}$ (onset at 8:00 a.m.) defined as low light (LL). Cells were cultured in sterile Provasoli's enriched F/2 seawater using Tropic Marin Classic artificial sea salts (Dr. Biener, Germany) with 16.6 g sea salt per 1 L medium. For all experiments, only cultures in the logarithmic growth phase were used, i.e., a chlorophyll *a* concentration of 1.5 mg L$^{-1}$ or a cell concentration of 3.5 million cells mL$^{-1}$ at maximum. Therefore, cultures were regularly diluted with fresh F/2 medium in order to avoid any nutrient limitation, which would strongly influence *Lhcx* expression[32]. Chlorophyll *a* concentration was determined as described in ref.[33]. Cell counts were obtained using a Multisizer 3 Coulter Counter (Beckman, USA). For long-term light stress experiments, cells were exposed to white light with an intensity of ~ 400 μmol photons m$^{-2}$ s$^{-1}$ for 24 h. Light intensities were measured with a spherical quantum sensor (US-SQS/L, Walz, Germany).

**Generation of TALEN and complementation constructs**. TALEN-knockout plasmids for Lhcx1 and Lhcx2 were generated following the procedure for creating TALEN-KO lines in *P. tricornutum* by Serif et al.[37]. This method relies on Golden Gate reactions, where digestion with TypeIIs restriction enzymes and ligation are carried out in one step[69]. For constructing the specific TALENs, we assembled the respective TALEN monomers into the respective backbone vectors available at Addgene. For Lhcx1 we constructed two TALEN pairs, targeting two different sites in the *Lhcx1* gene. For Lhcx2 we constructed one TALEN pair. All genomic target sites contained a thymine before the actual TALE recognition site. For each TALEN we assembled 18 monomers, each with a repeat variable di-residue (RVD), which recognizes one specific DNA base. The 19th monomer is only a half monomer and is included in the respective TALEN backbones[37]. Supplemental Table 1 indicates the RVDs of the respective TALEN constructs and the corresponding DNA target sequence as well as the backbone vector (available at Addgene), which contains the half monomer, the FokI nuclease-coding region as well as the antibiotics resistance cassette.

In order to complement/supplement our KO strains, *Lhcx1*, *Lhcx2*, *Lhcx3*, and *Lhcx4* were cloned inside a modified pPha-T1 vector pPTbsr, respectively, which instead of a Bleomycin resistance cassette contains a Blasticidin-S resistance cassette[70]. The *FcpA* promoter of this vector was removed via deletion PCR and replaced by two different *Lhcx* promoters: the genes designated to complement the x1KO background were cloned in between the *Lhcx1* promoter (starting 721 bases upstream to the *Lhcx1* gene start codon) and terminator (617 bases downstream of the *Lhcx1* stop codon), whereas for the genes designated to complement the x2KO strain the *Lhcx2* promoter (605 bp upstream of translation start of *Lhcx2*) and terminator (422 bp downstream of *Lhcx2* stop codon) was used. When the KO lines were complemented with their original gene (*Lhcx1* for x1KO, *Lhcx2* for x2KO), modified gene sequences using synonymous codons were used, in order to prevent the TALENs from cutting the newly introduced genes. Except for *Lhcx1*, all genes were cloned including their respective introns. In addition, the modified *Lhcx1* gene was also cloned in between the original *FcpA* promoter and terminator of the pPTbsr vector.

Sequences for the *Lhcx* genes of Pt4 are slightly different from those of the sequenced Pt1 strain (https://genome.jgi.doe.gov/Phatr2/Phatr2.home.html) (*Lhcx1* JGI ID: 27278; *Lhcx2* JGI ID: 44733; *Lhcx3* JGI ID: 44733; *Lhcx4* JGI ID: 38720). All gene, promoter and terminator sequences were obtained by amplifying genomic DNA with specific primers using Kapa Hifi DNA Polymerase (Roche, Switzerland)

and purification of the PCR product from agarose gels using Geneclean Turbo Kit (MP Biomedicals, Germany) except for the modified *Lhcx1* and *Lhcx2* sequences, which were ordered from BioCat (Germany). Correct amplification and vector integration were verified by sequencing (Microsynth AG, Switzerland). All sequences for the genes, promoters and terminators used in this study are provided in Supplementary Table 2. As an example, the transformation vector maps for *Lhcx3* complementing the x1KO and x2KO line, respectively, are depicted in Supplementary Fig. 19.

**Generation of mutated *P. tricornutum* cell lines**. Wild-type Pt4 cells were biolistically transformed with the TALEN bearing vectors according to[71]. Positive clones were selected on Zeocin (75 μg mL$^{-1}$; Invitrogen, USA) and Nourseothricin (150 μg mL$^{-1}$; ClonNAT, Werner Bioagents, Germany) containing solid medium plates containing 1.2 % Bacto Agar (BD, USA) in F/2 medium (16.6 g L$^{-1}$ sea salt). Pre-screening of the colonies was performed using an Imaging PAM system (Walz, Germany), searching for clones with altered NPQ characteristic. Further characterization included western blot, PCR, and DNA sequencing. Confirmed KO clones were spread on individual agar plates in a suitable dilution to obtain clones from single cells. Three out of these were reanalyzed for each knockout line and one of each KO line was used for all follow-up experiments. In addition, whole-genome sequencing was done on the three created x1KO lines. All initial KO clones observed were homogeneous, i.e., clones obtained from single cells from the initial clones all showed the same knockout phenotype.

Two obtained x1KO lines were complemented with a modified *Lhcx1* gene by a second biolistic transformation using Blasticidin-S (4 μg mL$^{-1}$) containing solid medium plates in low-salt F/2 medium (8.3 g L$^{-1}$ sea salt) for selection. In addition, the native *Lhcx2*, *Lhcx3*, and *Lhcx4* genes were transformed in the x1KO_1a line. The x2KO line was complemented with a modified *Lhcx2* gene, and also transformed with the native *Lhcx3* gene.

**DNA isolation and allele specific PCR**. Genomic DNA was isolated using the Nexttec 1step DNA isolation kit (Biozym, Germany) according to the manufacturer's instructions. A cell pellet corresponding to 10 mL of culture was used as starting material. The concentration of genomic DNA was measured with a Nanodrop 2000 (Thermo Fisher Scientific, Germany). In order to prove that both alleles were mutated, allele specific PCR was applied for the x2KO mutants only, as the *Lhcx1* gene of Pt4 does not contain allele specific differences in the TALEN-targeted region. The sequences of the two primer pairs for both alleles as well as all following primers are specified in Supplementary Table 3. The gene sequence for *Lhcx2* of Pt4 is depicted in Supplementary Fig. 20. Allele specific PCR, however, provided no PCR product in the x2KO line (Supplementary Fig. 6a). We additionally used the primer combination Lhcx2_prom-fw and Lhcx2_term-rev, which amplifies the *Lhcx2* promoter, *Lhcx2* gene and *Lhcx2* terminator in a length of ~ 1970 bp in wild-type. In the x2KO mutant, in one of the alleles a deletion of 918 bp occurred (Supplementary Fig. 6). The sequence of this mutated allele is indicated in Supplementary Fig. 21.

To screen the x1KO mutants and the complemented lines, we used different primer combinations. The primer pair Lhcx1_all-fw and Lhcx1_all-rev amplifies 102 bp at the 3′-end of the *Lhcx1* gene in the Pt4 strain, the x1KO strains and the Lhcx1 complemented lines. It served as a control. The primer pair Lhcx1_wild-type-fw and Lhcx1_wild-type-rev amplifies a 347 bp sequence only in the wild-type strain, whereas the primer pair Lhcx1_comp-fw and Lhcx1_wild-type-rev amplifies 262 bp only in the complemented x1KO lines. Note that for the two latter primer combinations only the forward primer was different, binding either to the wild-type sequence of *Lhcx1* or the modified sequence of the complemented *Lhcx1* gene with synonymous codon usage.

To screen the x1KO_1a + FcpA_x1-complemented mutants, the primer pair FcpA_Lhcx1-fw and FcpA_Lhcx1-rev amplifies a 479 bp fragment, including the *FcpA* promoter and the first base pairs of the modified *Lhcx1* gene. This combination of *FcpA* promoter and modified *Lhcx1* gene, as well as the modified TALEN-binding sites on which FcpA_Lhcx1-rev binds, are unique for the complementation construct and do not occur in wild type cells.

PCR was performed using HiDi polymerase according to the manufacturer's instructions for 30 cycles (MyPols, Germany), with either the primer pairs Lhcx1_wild-type-fw/Lhcx1_wild-type-rev and Lhcx1_all-fw/Lhcx1-all_rev or with Lhcx1_comp-fw/Lhcx1_wild-type-rev and Lhcx1_all-fw/Lhcx1-all_rev. PCR products were separated on 1% agarose gels.

**Whole-genome sequencing**. In order to identify the correct DNA sequence for both alleles of *Lhcx1* and *Lhcx2* in Pt4 and in order to verify the biallelic knockout of the *Lhcx1* gene in the x1KO lines, the genomic DNA of Pt4 and the three x1KO strains were isolated using the MasterPure DNA Purification Kit (Epicentre, USA) according to the manufacturer's instructions and sequenced by Illumina 125 bp paired end sequencing by GATC Eurofins (Germany). Quality control for raw reads was done using FastQC. Low quality reads (quality score ≤ 20) were trimmed by FASTQ/A Trimmer (http://hannonlab.cshl.edu/fastx_toolkit/index.html). As no reference genome for Pt4 existed so far, we first produced a new Pt4 reference genome, by assembling and aligning the trimmed reads of Pt4 with Bowtie2[72] against the available genome of Pt1 CCAP 1055/1 deposited in Ensembl, which is

an update of the previously deposited genome in JGI[36]. Then, the SAM files created by Bowtie2 were converted into BAM files using SAM tools[73] and a new consensus sequence for Pt4 was created by aid of the BCF tools pipeline[74]. Finally, the quality controlled and trimmed reads of the x1KO mutants were mapped against the reads of the new Pt4 reference genome by aid of Bowtie2.

**Fluorescence analyses**. Pre-screening of mutants was performed with an Aqua-Pen (PSI Instruments, Czech Republics) or an Imaging PAM (Walz, Germany). Fine fluorescence kinetics were recorded with a Dual-PAM or a Fluorescence Induction and Relaxation instrument (a mini-FIRe). For the Dual-PAM measurements, cells were concentrated to 10 µg chlorophyll $a$ mL$^{-1}$, and 4 mM NaHCO$_3$ was added. Before starting experiments, cells were acclimated to dim light for 30–45 min. Kinetic measurements were performed by exposing the cells to 10 min of actinic light with an intensity of 1700 µmol photons m$^{-2}$ s$^{-1}$, consisting of similar proportions of red and blue light photons, followed by 18 min of recovery conditions with 40 µmol photons m$^{-2}$ s$^{-1}$. Saturating flashes (8000 µmol photons m$^{-2}$ s$^{-1}$ red light, 800 ms) were applied every 30 s. For some experiment, DTT (500 mM final concentration) was added 5 min prior to fluorescence recording. To monitor acclimation under prolonged high light, cells were exposed for 130 min to 1700 µmol photons m$^{-2}$ s$^{-1}$, followed by 30 min recovery conditions with 40 µmol photons m$^{-2}$ s$^{-1}$. NPQ was calculated as Fm/Fm'–1. Here, we set Fm to the first light step Fm', as this value is usually higher than the Fm obtained for dark acclimated $P.$ $tricornutum$ cells.

The variable fluorescence signatures of minimal (Fo) and maximal (Fm) fluorescence, corresponding to states where all PSII reaction centers are open or closed, respectively, as well as the functional absorption cross-sections of PSII, were measured using a mini-FIRe[75]. Fluorescence induction was achieved with blue light emitting diodes (450 nm, 30-nm half bandwidth) and measured in the red light region (680 nm, 20 nm bandwidth). An 80 µs pulse, with peak optical power of 1 W/cm$^2$, ensured that all PSII reaction centers were reduced with a single turnover of PSII. From this single turnover protocol, parameters such as σPSII, electron transfer rates, NPQ and Y(NPQ) can be calculated[46]. For all strains, these parameters were measured in response to increasing actinic light in 15 steps with 1 min acclimation at each step (i.e., light curves, 0–800 µmol photons m$^{-2}$ s$^{-1}$ of blue light), where variable fluorescence (resulting in F' and Fm') was recorded at the end of each light step. In addition, after each light step, the actinic light was turned off for 1 s to allow for re-opening of reaction centers. This 1 s dark period was short enough to prevent NPQ relaxation[9,14] but did allow us to directly measure Fo', which is needed for the calculation of 1–qL. Electron transfer rates were calculated from E × σPSII'×(Fv'/Fm')/(Fv/Fm)[6]. Y(NPQ) and 1–qL were calculated following[9]. σPSII' was derived from F'→Fm' transition induced by an 80 µs single turnover flash[76]. Using the F'→Fm' transition allowed us to make more acquisitions, improving the signal-to-noise ratio, and reducing the error in the fitting procedure. A comparison of σPSII' and σPSII'$_{1s}$ (i.e., obtained after 1 s darkness) showed little difference in the values or the interpretation of our results (cf. Fig. 5, Supplementary Fig. 17).

Fluorescence lifetimes were measured using a custom-built picosecond lifetime fluorometer as described in ref. 46. We measured lifetimes for Fo and Fm levels in the dark, followed by measurements of Fo' and Fm' lifetimes during exposure to 850 µmol photons m$^{-2}$ s$^{-1}$ (blue light) for 10 min to induce NPQ. The fluorescence acquisition interval was 20 s, short enough to avoid changes in the state of the photosynthetic units. Average fluorescence lifetimes were calculated by fitting the collected data to a sum of three exponential decays, which were deconvoluted from the instrument response function[46].

**Pigment analysis**. Pigment extraction and HPLC separation followed the protocol established by[77]. In short, pigments of cells filtered on a 1.2 µm isopore filters (Millipore) were extracted with a mixture of 81% methanol/9% 0.2 M ammonium acetate/10% ethyl acetate (vol/vol/vol). After centrifugation, the supernatant was injected onto a calibrated Hitachi LaChrom Elite HPLC system equipped with a 10 °C-cooled autosampler and a Nucleosil 120-5 C18 column (Macherey-Nagel, Germany). Pigments were separated using a linear gradient system consisting of eluent A (90% methanol/10% 0.5 M ammonium acetate, vol/vol) and eluent B (90% methanol/10% ethyl acetate, vol/vol).

**Quantitative real-time PCR (qPCR)**. RNA extraction was performed using a combination of Peqgold RNAPure and Peqgold Total RNA Kit S-Line with on column DNA digest using the Peqgold DNAse I Digest kit (VWR, Germany). cDNA synthesis was performed by means of the Primescript kit (Takara Bio Europe, France). RNA was extracted from LL grown cultures in the exponential growth phase. In order to assess $Lhcx$ transcript level of light-stressed cells, the different strains were concentrated to 2 µg mL$^{-1}$ chlorophyll $a$, treated with 0.8 mM NaHCO$_3$ and exposed for 2 hours to ~700 µmol photons m$^{-2}$ s$^{-1}$ white light in a glass cylinder under constant water-cooling and stirring at 20 °C. For the qPCR analyses, the reference gene was $18S\_rRNA$ (Ensembl Gene ID: EPr-Phatr3G00000013183) as it is well-suited for both low and high-light acclimated $P.$ $tricornutum$ cultures[33]. Primer sequences for $Lhcx1$, $Lhcx3$, $Lhcx4$, and $18S$ were the same as in ref. 33 and are indicated in Supplementary Table 3. For $Lhcx2$ we used Lhcx2_qPCR_fw and Lhcx2_qPCR_rev (Supplementary Table 3). This primer

pair recognizes both alleles, in contrast to that used by[33]. qPCR was run on a 7500 Fast RT-PCR system (Applied Biosystems, USA). Each strain was measured in biological triplicates, and on top each gene per sample was measured three times. Cycle threshold values and gene amplification efficiencies were obtained by utilizing PCR Miner 4.0[78]. Relative transcript levels were calculated according to[79].

**Western blot**. Protein extraction and separation followed the protocol described by ref. 67, but using 14% lithium dodecyl sulfate-polyacrylamide gel electrophoresis for protein separation. Samples corresponding to an amount of 1 µg chlorophyll $a$ were loaded on the gel. Proteins were blotted on a Bio-Trace PVDF (Pall Corporation, USA) or an Amersham Protran nitrocellulose membrane (GE Healthcare, GBR) using the semidry blotting technique by means of a Biorad Trans-Blot Turbo system (Hercules, USA). Previously, either an anti-Lhcsr, raised against a recombinant $Chlamydomonas$ Lhcsr protein, had been used to quantify Lhcx protein expression in $P.$ $tricornutum$[18,32–34] or it had been the anti-FCP6 raised against a specific C-terminal peptide of the FCP6 protein in $Cyclotella$ $cryptica$[30,35]. We designed a new polyclonal anti-$P.$ $tricornutum$-Lhcx (manufactured by Agrisera, Sweden) raised against the peptide MAQELVNGKGILEHL, because 14 (Lhcx1, Lhcx3), 13 (Lhcx2), and 12 (Lhcx4) out of these 15 amino acids are contained specifically in the C-terminus of the respective Lhcx proteins in $P.$ $tricornutum$, thus making this antibody superior in detecting the different Lhcx proteins with a similar affinity. Anti-Lhcx was applied in 1:10,000 dilution overnight. Anti-Rubisco (AS03 037, Agrisera, Sweden; 1:10,000 dilution) served as a loading control. After binding of the secondary antibody for 1 h (goat anti-rabbit IgG, (H&L) HRP conjugated, AS09 602, Agrisera, Sweden; 1:10,000 dilution for Lhcx, 1:20,000 for Rubisco), signals were detected using Roti-Lumin Plus (Carl Roth, Germany) in an Odyssey FC Imaging System (LI-COR, USA).

**Statistics**. Significance with a $P$ value ≤ 0.05 was determined with a two-tailed unpaired Student's $t$ test calculated with SigmaPlot 12 on biological replicates as indicated in the respective legends. For gene expression we used the Pairwise Fixed Reallocation Randomization Test performed by REST2006 according to[80] with 2000 randomizations.

**Reporting summary**. Further information on research design is available in the Nature Research Reporting Summary linked to this article.

## Data availability
Source data for the figures and supplementary figures are provided with the article. Any further data related to the article is available from the corresponding author upon reasonable request. Whole-genome sequencing reads for Pt4, x1KO_1a, x1KO_1b, and x1KO_2 are deposited at the European Nucleotide archive (ENA) under the following accession code: "PRJEB33825".

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

## Acknowledgements

We thank Doris Ballert for biolistic transformation of *Phaeodactylum tricornutum*, Annette Ramsperger, and Kristoffer Weißert for helping steps during the TALEN- cloning procedure, Teresa Hagen for assistance in selection of positive clones and Angelika Eckert for help during the growth experiments. We also thank Doug Campbell for very useful discussion. This research was supported by the DFG (grant KR 1661/8-2 to P.G.K. and grant LE 3358/3-1 to B.L.), the Baden-Württemberg Stiftung (Elite program to B.L.), the CNRS-Centre National de la Recherche Scientifique France (to J.L.), the NSERC-Natural Sciences and Engineering Research Council of Canada (to J.L.), the Swiss National Science Foundation (Grant: 31003A_172964 to JH), the NASA Ocean Biology and Biogeochemistry Program (to M.G. and P.G.F.), a Marie Curie Zukunftskolleg Incoming Fellowship (grant no. 291784, to B.L.), a Zukunftskolleg Interim grant (to B.L.) and a Zukunftskolleg Interdisciplinary Collaborative Project (to B.L.).

## Author contributions

J.M.B., J.S., J.L., A.F., P.G.K., M.Y.G., P.G.F., and B.L. designed experiments. J.M.B., C.R.B., M.S., M.H., and B.L. produced the mutants. J.M.B., C.R.B., M.S., and B.L. screened the mutants via PCR, western blot, growth, and NPQ experiments. J.H., C.R.B., and B.L. analyzed the whole-genome sequencing data. J.M.B. and B.L. performed in-depth NPQ, HPLC, and gene expression analyses. J.S., J.M.B., and B.L. recorded FIRe and fluorescence lifetime data. J.S., M.Y.G., P.G.F., J.M.B., and B.L. analyzed the FIRe and fluorescence lifetime data. J.M.B., J.S., A.F., J.L., M.Y.G., P.G.K., P.G.F., and B.L. wrote the article.

## Additional information

**Competing interests:** The authors declare no competing interests.

