## [Peer Review File · Nature Communications]

Reviewers' comments:

Reviewer #1 (Remarks to the Author):

The manuscript by unknown authors deals with the function of LHCX proteins of *Phaeodactylum tricornutum* in inducing NPQ and reducing antenna size of PSII, the latter being a debated question in the recent literature. Authors used genome editing and over-expression methods in order to build a range of genotypes which widely diversified capacity for NPQ. Authors convincingly show that LHCX1,2 and 3 have a similar effect on NPQ activity, their differential efficiency being due to the difference in expression level. LHCX4 is different and its function remains to be studied. The above claim is in part confirmatory with previous work by Taddei and co-workers and yet new data add to the knowledge in the field. The second claim is that the relation between energy dissipation and change in antenna size is maintained through a large range of NPQ values. This is also well documented by several measurements which are well performed and interpreted. This is in clear contrast with previous work by Ruban and co-workers in both plants and diatoms, which has recently risen interest and criticism. In this respect the present work is of general interest: larger than the diatom work. Also very interesting is the conclusion that NPQ involves disconnection of the outer antenna from PSII core. The conclusion appears to be well based on experiments. It is odd, however, that the work by Betterle et al, J. Biol. Chem. 2009, structurally documenting the disconnection of outer antenna from PSII core is not cited in this context since it is relevant and strongly supports the author's view. Although the authors do aim to study PSII, authors disregard the effect on PSI antenna size, which has been documented in both algae and mosses and might be relevant on LHCX4 function.

Reviewer #2 (Remarks to the Author):

The manuscript 'LhcX proteins provide photoprotection via thermal dissipation of absorbed light in the diatom *Phaeodactylum tricornutum*' describes the role of the different members of the LHCX family in protecting this diatom from light stress. This work represents an important technical achievement: several mutant lines have been generated to assess the different roles of the different LHCX.

However, I am really disappointed by the scientific focus of this manuscript. While the authors mention major open questions concerning NPQ in diatoms (the existence of several quenching sites, the localisation of LHCX proteins in the photosynthetic complexes, the modulation of the NPQ capacity in changing environmental conditions via different LHCX isoforms) they basically do not address them. Instead, they use their line as a 'tool' to corroborate previous knowledge on NPQ: the role of LHCX proteins in modulating the NPQ amplitude (for a given deepoxidation state), the

complementary role of LHCX1, LHCX2 and LHCX3, the changes in the PSII absorption cross section upon establishment of NPQ.

This is really a pity, because this choice largely diminishes the interest of the manuscript.

Reviewer #3 (Remarks to the Author):

The authors present a well designed set of experiments supporting a well argued interpretation of non-photochemical dissipation of excitation in the model diatom *Phaeodactylum*.

The authors show a clear regression between induction of a rapid 'qE' phase of non-photochemical quenching and progressive down-regulation of the effective absorbance cross section of PSII photochemistry.

A slower, minor phase does not, in contrast, contribute to down regulation of σ_{PSII} .

They use a set of mutants to show that 3 of the 4 LhcX isoforms in this diatom can support accumulation of qE, but only in conjunction with accumulation of the key xanthophyll pigment Dt.

They discuss these findings and reconcile them with some previous studies using alternate approaches to narrow the field of plausible models.

Diatom non-photochemical quenching has generated much heat and some light, and this contribution goes a long way to clearing the conceptual space.

I congratulate the authors and offer a few minor suggestions for data plotting and interpretation.

with best regards, Doug Campbell

ps. sorry to be slow, this arrived at a chaotic time.

Abstract:

Good.

Main Text Introduction:

Results & Discussion

Line 146: It would be good to specify that this σ_{PSII} is for blue light.

Line 180: To 'bound' NPQ between 0 and 1 (not 'bind')

Line 242:

"Our measurements of a significantly reduced σ_{PSII} under qE conditions are fully

243 consistent with the proposal that qE reduces the effective excitation diffusion length, i.e. the distance an

244 exciton can travel until it reaches an open PSII reaction center⁶⁵"

I think, rather:

""Our measurements of a significantly decreased σ_{PSII} under qE conditions are fully

consistent with the proposal that qE shortens the time an exciton can travel before being lost as heat, and thus the effective excitation diffusion length for an exciton to travel to reaches an open PSII reaction center before being lost to heat⁶⁵"

Figure 1:

The critical symbology distinguishing strains is only visible at 200% on my screen.

Figure 3:

Very nice; the slower accumulating phase of NPQ is independent of DTT and LhcX knockouts.

Figure 4:

Would be more helpful to me if there was a single LL column for all measured strains, and a single HL column for all measured strains.

This would facilitate the comparison of growth light effect for each strain; whereas currently the left right column comparisons are between arbitrary pairs of strains grown under LL (A) or HL (B)

Fig. 5:

Lovely plot, but I am struggling to understand the justification for:

Line 187 &

"Data points where an increase of $Y(NPQ)$ did not lead to a

further down-regulation of σ_{PSII} are not included in the regression calculation, but indicated in light grey. This was determined by calculating the percent change of each σ_{PSII} from its previous light step.

If σ_{PSII} decreased by less than 5% of the total measured decrease for that curve while $Y(NPQ)$ increased, it was omitted."

Why?

Eyeballing the data it would have little effect on the slope, and perhaps an upward offset in the Y intercept and a broadening of the confidence interval.

I am also struggling to understand the straight confidence interval lines, normally they are diverging curves at each extreme, to account for the increasing influence of slope uncertainty at extremes.

Most critically, what is the conceptual justification for excluding data points that do not reflect the expected pattern?

If the excluded points were included, I think the residual σ_{PSII} at $Y(NPQ) = 1$ would be very close to the parallel estimate Xu et al. 2018 of $167 \text{ A}^2 \text{ PSII-1}$ (which would be a quite astonishing confirmation of the pattern, given the differences in strains, growth conditions and instrumentation)

References:

Xu, K., Grant-Burt, J. L., Donaher, N. & Campbell, D. A. Connectivity among Photosystem II

694 centers in phytoplankters: Patterns and responses. *BBA-Bioenergetics* 1858, 459-474 (2017).

I think the authors mean instead the subsequent paper, which is where we did the regressions of σ_{PSII} vs. YNPQ (Fig. 4) (cited as Ref. 11 in the list)

Xu, K. et al. Phytoplankton σ_{PSII} and Excitation Dissipation; Implications for Estimates of Primary Productivity. *Front. Mar. Sci.* 5, (2018).

We thank the reviewers for their efforts and constructive remarks which helped us improve our manuscript. The specific responses to the individual comments follow below.

Reviewer #1 (Remarks to the Author):

“The manuscript by unknown authors deals with the function of LHCX proteins of *Phaeodactylum tricornutum* in inducing NPQ and reducing antenna size of PSII, the latter being a debated question in the recent literature. Authors used genome editing and over-expression methods in order to build a range of genotypes with widely diversified capacity for NPQ. Authors convincingly show that LHCX1,2 and 3 have a similar effect on NPQ activity, their differential efficiency being due to the difference in expression level. LHCX4 is different and its function remains to be studied. The above claim is in part confirmatory with previous work by Taddei and co-workers and yet new data add to the knowledge in the field. The second claim is that the relation between energy dissipation and change in antenna size is maintained through a large range of NPQ values. This is also well documented by several measurements which are well performed and interpreted. This is in clear contrast with previous work by Ruban and co-workers in both plants and diatoms, which has recently raised interest and criticism. In this respect the present work is of general interest: larger than the diatom work. Also very interesting is the conclusion that NPQ involves disconnection of the outer antenna from PSII core. The conclusion appears to be well based on experiments. It is odd, however, that the work by Betterle et al, J.Biol. Chem.2009, structurally documenting the disconnection of outer antenna from PSII core is not cited in this context since it is relevant and strongly supports the author's view. Although the authors do aim to study PSII, authors disregard the effect on PSI antenna size, which has been documented in both algae and mosses and might be relevant on LHCX4 function.”

We thank the reviewer for the well-made summary of our major findings and for the estimation of our work. Regarding Betterle et al. 2009, the reviewer is right, it fits nicely to our described data and is now included in the discussion (Line 302 of the manuscript with tracked changes: “Such a mechanism has been identified as one compound of NPQ in *A. thaliana* using electron microscopy⁷¹”). Also, the remark regarding cross section of PSI is valuable. NPQ/thermal dissipation is usually studied referring on PSII fluorescence which was the focus of this manuscript but in future work we will tackle also the question of a putatively variable cross section in PSI also with respect of LhcX4.

Reviewer #2 (Remarks to the Author):

“The manuscript ‘LhcX proteins provide photoprotection via thermal dissipation of absorbed light in the diatoms *Phaeodactylum tricornutum*’ describes the role of the different members of the LHCX family in protecting this diatom from light stress. This work represents an important technical achievement: several mutant lines have been generated to assess the different roles of the different LHCX. However, I am really disappointed by the scientific focus of this manuscript. While the authors mention major open questions concerning NPQ in diatoms (the existence of several quenching sites, the localisation of LHCX proteins in the photosynthetic complexes, the modulation of the NPQ capacity in changing environmental conditions via different LHCX isoforms) they basically do not address them. Instead, they use their line as a ‘tool’ to corroborate previous knowledge on NPQ: the role of LHCX proteins in modulating the NPQ amplitude (for a given deepoxidation state), the complementary role of LHCX1, LHCX2 and LHCX3, the changes in the PSII absorption cross section upon establishment of NPQ. This is really a pity, because this choice largely diminishes the interest of the manuscript.”

Reviewer #3 (Remarks to the Author):

“The authors present a well designed set of experiments supporting a well argued interpretation of non-photochemical dissipation of excitation in the model diatom *Phaeodactylum*. The authors show a clear regression between induction of a rapid 'qE' phase of non-photochemical quenching and progressive down-regulation of the effective absorbance cross section of PSII photochemistry.

A slower, minor phase does not, in contrast, contribute to down regulation of sigmaPSII. They use a set of mutants to show that 3 of the 4 LhcX isoforms in this diatom can support accumulation of qE, but only in conjunction with accumulation of the key xanthophyll pigment Dt. They discuss these findings and reconcile them with some previous studies using alternate approaches to narrow the field of plausible models.

Diatom non-photochemical quenching has generated much heat and some light, and this contribution goes a long way to clearing the conceptual space.

I congratulate the authors and offer a few minor suggestions for data plotting and interpretation. with best regards, Doug Campbell

ps. sorry to be slow, this arrived at a chaotic time.“

Thanks, Doug, for this nice summary and also for the constructive comments below.

Abstract:

Good.

Results & Discussion

Line 146: It would be good to specify that this sigmaPSII is for blue light.

We had specified this in the discussion and in the methods, but now specified this also at the requested place (L153, underlined the changes): “ σ PSII values of low light grown strains, measured in the dark, were between 500 and 550 $\text{\AA}^2 \text{PSII}^{-1}$ upon blue light (450 nm) exposure, typical for *P. tricornutum*⁵⁰.”

Line 180: To 'bound' NPQ between 0 and 1 (not 'bind')

Done.

Line 242:

"Our measurements of a significantly reduced σ PSII under qE conditions are fully consistent with the proposal that qE reduces the effective excitation diffusion length, i.e. the distance an exciton can travel until it reaches an open PSII reaction center⁶⁵"

I think, rather:

""Our measurements of a significantly decreased σ PSII under qE conditions are fully consistent with the proposal that qE shortens the time an exciton can travel before being lost as heat, and thus the effective excitation diffusion length for an exciton to travel to reaches an open PSII reaction center before being lost to heat⁶⁵"

We changed to (L267): "Our measurements of a significantly decreased σ PSII under qE conditions are fully consistent with the proposal that qE shortens the time an exciton can travel before being thermally dissipated, and thus decreases the effective excitation diffusion length for an exciton to reach an open PSII reaction center⁶⁷."

Figure

1:

The critical symbology distinguishing strains is only visible at 200% on my screen.

We increased symbol size and line width – should be better visible now.

Figure 3:

Very nice; the slower accumulating phase of NPQ is independent of DTT and LhcX knockouts.

Exactly. And this is also why we excluded this phase for our sigma versus Y(NPQ) correlation in Fig. 5. From all what we know so far, this phase is no qE, but is rather related to qI processes in the reaction centers (see also Kuzminov and Gorbunov 2016). We now added two specifications:

L98: "While the x1KO line showed only a slight linear increase in NPQ which did not relax under LL conditions and may rather be related to photoinhibition processes (qI), the x1KO+x2 and x1KO+x3 supplemented lines recovered qE capacity (Fig. 3a; Suppl. Fig. 4)".

L109: "Hence, Dt can only confer qE in the presence of LhcX1/2/3 proteins, and *vice versa*, and the slower NPQ phase, observed at prolonged high light intensities, is independent of both compounds".

This compound may also be one reason why there has been so much confusion in the past about two different regulated quenching sites in *P. tricornutum* while there has been overwhelming evidence for a 1:1 relation between xanthophyll cycle activity and qE. Our data clearly demonstrate that there is only one rapidly regulated quenching site which is based on both, LhcX and xanthophyll cycle activity, as well as detachment of antennae. This is also discussed in the manuscript, see also point 3 in response to reviewer #2.

Figure 4:

Would be more helpful to me if there was a single LL column for all measured strains, and a single HL column for all measured strains.

This would facilitate the comparison of growth light effect for each strain; whereas currently the left right column comparisons are between arbitrary pairs of strains grown under LL (A) or HL (B)

Fig. 4 now contains a left column with LL data and a right column with corresponding HL data.

Fig. 5:

Lovely plot, but I am struggling to understand the justification for:

Line 187 &

"Data points where an increase of $Y(NPQ)$ did not lead to a further down-regulation of $\sigma PSII$ are not included in the regression calculation, but indicated in light grey. This was determined by calculating the percent change of each $\sigma PSII$ from its previous light step.

If $\sigma PSII$ decreased by less than 5% of the total measured decrease for that curve while $Y(NPQ)$ increased, it was omitted."

Why?

Eyeballing the data it would have little effect on the slope, and perhaps an upward offset in the Y intercept and a broadening of the confidence interval.

I am also struggling to understand the straight confidence interval lines, normally they are diverging curves at each extreme, to account for the increasing influence of slope uncertainty at extremes.

Most critically, what is the conceptual justification for excluding data points that do not reflect the expected pattern?

If the excluded points were included, I think the residual $\sigma PSII$ at $Y(NPQ) = 1$ would be very close to the parallel estimate Xu et al. 2018 of $167 A^2 PSII-1$ (which would be a quite astonishing confirmation of the pattern, given the differences in strains, growth conditions and instrumentation)

The confidence intervals are diverging towards the maximum NPQ due to the higher uncertainty here, but as the fit is very good, this is only visible when extending the x-axis. This is shown below in a graph.

The aim of excluding data points was to show only the correlation between qE and decreases in $\sigma PSII$. $NPQ/(Y(NPQ))$ determination cannot distinguish between qE and qI , but our data clearly show that only the rapid part of NPQ establishment is based on a concerted action of the xanthophyll cycle and Lhcx proteins, which simultaneously leads to a lowering of $\sigma PSII$ - which we also used as a new feature to determine qE . NPQ occurring at prolonged light exposure and higher light intensities is independent of all three compounds and rather reflects qI . Thus, in order to investigate the influence of Lhcx proteins on qE and $\sigma PSII$, these latter NPQ processes need to be filtered out, to avoid an artificial distortion of the regression upwards. Probably, here we were not clear enough in the manuscript. In order to make this scientifically valid and not just based on our visual intuition, we calculated those values where no substantial decrease of $\sigma PSII$ was obtained as indicated in the manuscript. In addition, we excluded data points above $600 \mu mol m^{-2} s^{-1}$, as at those light intensities the establishment of qE was always terminated and $\sigma PSII$ values became noisy due to a low PSII yield and a resulting less solid $\sigma PSII$ calculation. So far, in the manuscript we had:

L149: *"We do not consider the slight linear increase of NPQ at higher light intensities as qE. We observed this increase in all strains and it was independent of the presence of Lhcx proteins or of a functional Dd de-epoxidation."*

L159: *"Further increased light intensities, which induced a slight linear increase of NPQ capacity not related to qE (see above), did not further down-regulate $\sigma PSII$."*

L185: *"As for LL grown cells, the rapid decline of $\sigma PSII$ up to light intensities of 250-400 $\mu mol photons m^{-2} s^{-1}$ well correlated with the rapid development of qE, with the exception of the decline of $\sigma PSII$ under very weak light intensities in the beginning (Fig. 4b). The weaker and linear increase of NPQ under further increased light intensities did not have any effect on $\sigma PSII$."*

L202: *"We plotted all $\sigma PSII$ values against $Y(NPQ)$ for both LL and high light grown strains together, excluding those points where $Y(NPQ)$ was absent or did not further downregulate $\sigma PSII$ (see details legend Fig. 5)."*

L256: *"Our results also showed that the slow phase of NPQ, which is activated under prolonged exposure to higher light levels, has no effect on $\sigma PSII$. A similar slow NPQ phase with no reduction in $\sigma PSII$ is observed in the mutants lacking qE ($x1KO$ and $x1KO+x4$) as well as in all strains treated with DTT (which inhibits the formation of Dt). These results suggest that the slow phase of NPQ is related to photoinhibitory damage to the PSII reaction center (qI)."*

L262: *"The combination of NPQ and $\sigma PSII$ measurements in a variety of mutants with different NPQ extents allowed us to better characterize multiple NPQ processes that are otherwise hardly distinguishable^{3,4}. Accordingly, the most rapid NPQ process (qE) can be easily identified by a pronounced reduction in $\sigma PSII$ upon exposure to high light."*

We now added in the legend of Fig. 5 (underlined):

“In order to discern Lhcx and Dt dependent qE processes from qI processes, data points where an increase of Y(NPQ) did not lead to a further down-regulation of σ_{PSII} are not included in the regression calculation, but indicated in light grey.”

Additionally, following phrase L202 we added (L204):

“These points represent σ_{PSII} values measured when NPQ shifted from the fast qE component to a slower component and had no additional effect on σ_{PSII} (see discussion).”

Finally, we added (L284):

“As we obtained a linear correlation between the extent of Y(NPQ) and the decrease in σ_{PSII} driven directly by qE (Fig. 5, Suppl. Fig. 16), we can use the regression equation to calculate σ_{PSII} for a theoretical maximal Y(NPQ) of 1, i.e. where all absorbed energy is dissipated as heat.”

To highlight the effect of including different data point sets on the regression line and for the reviewers’ interest, we below show the resulting different plots. Still, we would like to keep Fig. 5 in the manuscript as it is, as it best highlights the effect of regulated thermal dissipation, based on the concerted action of Lhcx and the xanthophyll cycle, on the reduction of the PSII cross section and the functional implications of the remaining sigma at Y(NPQ) = 1 for the mechanistic understanding of qE.

Regression of sigma vs Y(NPQ) with all data points included. This means, an increasing Y(NPQ) without any effect on sigma decrease is also depicted (which provides horizontal data lines), and is taken into account for calculation of the regression line. This leads to an artificial upwards offset at Y(NPQ)=1 to a sigma of 141 \AA^2 . Because of the color coding, the artificial effect is best visible by looking at x1KO+x2a LL (filled green diamonds) and x2KO+x3 LL (filled purple triangles).

Regression of sigma vs Y(NPQ) with all data points included, except those where sigma decreased by less than 5%. This means, some data points at light intensities above 600 μE are also included, which leads to some Y(NPQ) e.g. in the x1KO 1a line (filled orange squares), which is, however, related to qL. The remaining sigma of 119 \AA^2 at Y(NPQ)=1 is still very close to the calculated one of the monomeric PSII core (112 \AA^2).

Regression of sigma vs Y(NPQ) with all data points below 600 μE included. This means, an increasing Y(NPQ) without any effect on sigma decrease is also depicted (which provides horizontal data lines), and is taken into account for calculation of the regression line. This leads to an artificial upwards offset at Y(NPQ)=1 to a sigma of 125 \AA^2 . Because of the color coding, the artificial effect is again best visible by looking at x1KO+x2a LL (filled green diamonds) and x2KO+x3 LL (filled purple triangles).

Prolongation of the regression line and the confidence intervals to a hypothetical value of $Y(\text{NPQ}) = 10$ and a corresponding elongation of the y-axis to a hypothetical cross section of -4000 A^2 . Now the spreading of the CI intervals, as argued by reviewer #3, becomes visible.

References:

Xu, K., Grant-Burt, J. L., Donaher, N. & Campbell, D. A. Connectivity among Photosystem II 694 centers in phytoplankters: Patterns and responses. *BBA-Bioenergetics* 1858, 459-474 (2017).

I think the authors mean instead the subsequent paper, which is where we did the regressions of σ_{PSII} vs. $Y(\text{NPQ})$ (Fig. 4) (cited as Ref. 11 in the list)

Xu, K. et al. Phytoplankton σ_{PSII} and Excitation Dissipation; Implications for Estimates of Primary Productivity. *Front. Mar. Sci.* 5, (2018).

To our best knowledge, whenever we referred to the linear correlation of σ vs $Y(\text{NPQ})$, we used the correct reference (Xu et al. *Front. Mar. Sci.* 5 (2018), *number 11 in the list*). We used Xu et al. *BBA-Bioenergetics* 1858 (2017) (*number 60 in the list*) only once, and this in a different context (Line 240): "Instead, they show a slight increase in σ_{PSII} , as expected when PSII reaction centers become progressively closed, causing excitons to jump from one PSII unit to another, in search for an open PSII center⁵⁸⁻⁶⁰."

And here, we believe the reference is well justified. We refer to Fig. 13G as well as the phrases on page 465, chapter 4.2 in Xu et al. 2017: "When Joliot and Joliot [7] proposed the notion of ρ , they concluded that an exciton that visits a closed PSII RC can be redirected to an open PSII RC, so that the trapping cross section of remaining open PSII RC increases as their neighbors become closed. Therefore, σ_{PSII} of remaining open PSII RC should increase when neighbors become closed by actinic light."

REVIEWERS' COMMENTS:

Reviewer #1 (Remarks to the Author):

The manuscript has been substantially improved during revision. The measurements required were performed with exception to those that required unreasonable amount of time. The discussion is now better defined and the major results have been highlighted. I think the manuscript is suitable for publication.

Reviewer #3 (Remarks to the Author):

The authors have fully addressed my minor comments and I congratulate them on a rigorous and well argued manuscript on an important topic.